# Marginal versus Standard Donors in Heart Transplantation: Proper Selection Means Heart Transplant Benefit

**DOI:** 10.3390/jcm11092665

**Published:** 2022-05-09

**Authors:** Olimpia Bifulco, Tomaso Bottio, Raphael Caraffa, Massimiliano Carrozzini, Alvise Guariento, Jonida Bejko, Marny Fedrigo, Chiara Castellani, Giuseppe Toscano, Giulia Lorenzoni, Vincenzo Tarzia, Dario Gregori, Massimo Cardillo, Francesca Puoti, Giuseppe Feltrin, Annalisa Angelini, Gino Gerosa

**Affiliations:** 1Department of Cardiac, Thoracic, Vascular Sciences and Public Health, University of Padua, 35128 Padua, Italy; bif.oly92@gmail.com (O.B.); raphael.caraffa@gmail.com (R.C.); alvise.guariento@unipd.it (A.G.); marny.fedrigo@gmail.com (M.F.); chiara.castellani@unipd.it (C.C.); giuseppe.toscano@aopd.veneto.it (G.T.); giulia.lorenzoni@unipd.it (G.L.); v.tarzia@gmail.com (V.T.); dario.gregori@unipd.it (D.G.); annalisa.angelini@unipd.it (A.A.); gino.gerosa@unipd.it (G.G.); 2Cardiothoracovascular Department, Niguarda Hospital, 20162 Milan, Italy; massimiliano.carrozzini@gmail.com; 3Vascular Unit, Portogruaro Hospital, 30026 Portogruaro, Italy; jonidabg@gmail.com; 4Unit of Biostatistics, Epidemiology and Public Health, Department of Cardiac, Thoracic, Vascular Sciences and Public Health, University of Padua, 35128 Padua, Italy; 5Italian National Transplant Centre, National Institute of Health, 00162 Rome, Italy; massimo.cardillo@iss.it (M.C.); francesca.puoti@iss.it (F.P.); 6Regional Transplant Centre, 35128 Padua, Italy; giuseppe.feltrin@aopd.veneto.it

**Keywords:** heart marginal donors, in-hospital mortality, mid-term survival, standard donors and marginal donors’ comparison

## Abstract

Background: In this study, we assessed the mid-term outcomes of patients who received a heart donation from a marginal donor (MD), and compared them with those who received an organ from a standard donor (SD). Methods: All patients who underwent HTx between January 2012 and December 2020 were enrolled at a single institution. The primary endpoints were early and long-term survival of MD recipients. Risk factors for primary graft failure (PGF) and mortality in MD recipients were also analyzed. The secondary endpoint was the comparison of survival of MD versus SD recipients. Results: In total, 238 patients underwent HTx, 64 (26.9%) of whom received an organ from an MD. Hospital mortality in the MD recipient cohort was 23%, with an estimated 1 and 5-year survival of 70% (59.2–82.7) and 68.1% (57.1–81), respectively. A multivariate analysis in MD recipients showed that decreased renal function and increased inotropic support of recipients were associated with higher mortality (*p* = 0.04 and *p* = 0.03). Cold ischemic time (*p* = 0.03) and increased donor inotropic support (*p* = 0.04) were independent risk factors for PGF. Overall survival was higher in SD than MD (85% vs. 68% at 5 years, log-rank = 0.008). However, risk-adjusted mortality (*p* = 0.2) and 5-year conditional survival (log-rank = 0.6) were comparable. Conclusions: Selected MDs are a valuable resource for expanding the cardiac donor pool, showing promising results. The use of MDs after prolonged ischemic times, increased inotropic support of the MD or the recipient and decreased renal function are associated with worse outcomes.

## 1. Introduction

Heart transplantation (HTx) is the gold standard therapy in end-stage heart failure, although it is limited by the shortage of available donor organs [1]. In 2020, a death rate of 3–5% on the waiting list for heart transplant has been estimated with an average waiting time of about 4 years [2].

Considering the growing demand for cardiac organs, despite recent advances in mechanical circulatory support, the inclusion of selected marginal donors (MDs) has been proposed [3,4,5,6,7,8,9]. Controversial results have been reported on posttransplant outcomes following an MD donation, with relatively short-term data [7]. The main concerns are related to increased susceptibility to primary graft failure (PGF), coronary graft vasculopathy and high perioperative mortality [7].

The aim of this study (retrospective, observational) was to analyze the mid-term outcomes of recipients from MDs in a single-center experience, comparing them to those of patients receiving a standard donation (SD), and identifying the risk factors for an improper matching. In particular, we investigated early and mid-term survival of MD recipients and compared the survival of MD versus SD recipients.

## 2. Materials and Methods

### 2.1. Study Population

This was a retrospective, observational, single-center study of all consecutive patients who received an HTx at the University of Padua (Italy) since the start of the MD program, between January 2012 and December 2020. Patients who received organs from SDs and MDs were included (Table 1). All pediatric patients (<18 years of age), multiorgan donors and cardiac retransplants were excluded.

The medical records of all patients were reviewed. Anonymity and professional confidentiality were respected. Every reasonable effort was made to obtain written informed consent to participate in this study. In particular, the use of data for scientific and research purposes was already included in the written informed consent used. The local Institutional Review Board (Azienda Ospedaliera Università, Padua, Italy) approved the study design, the consent, and the review of the data (IBR number 48421; 23 September 2021). The uses of collected information and statistical analysis exclusively for scientific purposes were also granted by the Scientific Committee of the Italian National Transplant Network on 11 February 2021.

### 2.2. Outcomes and Definitions

The primary endpoints were early and mid-term survival of MD recipients. The secondary endpoint was the comparison of survival of MD versus SD recipients. Risk factors for primary graft failure (PGF) and mortality (30-day mortality, in-hospital mortality) in MD recipients were also analyzed.

MDs were defined with the following criteria: age over 60 years; reduced left ventricular performance (ejection fraction between 40–50%); left ventricular hypertrophy (septal thickness > 14 mm on echocardiographic evaluation); focal lesion of the coronary artery; significant valvular heart disease [10]. Primary graft failure (PGF) was defined according to the recent ISHLT consensus statement [11]. Acute cellular and antibody-mediated rejections were classified according to the guidelines of the International Society for Heart and Lung Transplantation (ISHLT) and treated in cases of grade ≥ 2 [12]. Coronary graft vasculopathy (CAV) was defined according to the ISHLT guidelines [13].

During the study period, the HTx waiting list in our Organ Procurement Organization was structured as follows.

Status 1:
a.mechanical circulatory support (MCS) due to acute haemodynamic deterioration; RVAD or biventricular assist device (BiVAD); LVAD with device-related complications; total artificial heart (TAH); intra-aortic balloon pump (IABP); extracorporeal membrane oxygenation (ECMO);b.mechanical ventilation.
Status 2:
a.uncomplicated LVAD; continuous inotrope infusion; patients with implantable cardioverter defibrillator (ICD) and malignant relapsing ventricular arrhythmias;b.outpatients, not included in the categories listed above.
Status 3: temporarily inactive.

Patients in Status 1 were eligible to enter into the national High-Urgency Program, where they were prioritized at a national level.

### 2.3. Cardiac Transplantation Protocol

All grafts were retrieved from brain-dead beating heart donors. Laboratory tests, transthoracic echocardiography and chest X-rays were used for the evaluation of the donors. A coronary angiography was performed whenever possible. Cardioplegic arrest was achieved with antegrade crystalloid solution at 4 °C (Celsior^®^). During transport, the graft was protected with topical hypothermia (ice-cold Ringer’s solution). All heart transplants were orthotopic, with bicaval anastomosis.

After transplantation, the induction of immunosuppressive therapy with intravenous antithymocyte globulin (0.1 mg/kg/h for three days) and methylprednisolone (1 g intraoperative, then 125 mg/8 h for two days) was initiated. Routine maintenance immunosuppressive drugs consisted of calcineurin inhibitors, mainly cyclosporine (1–2 mg/kg/12 h aiming 150–250 ng/mL), prednisone (2.5–7.5 mg per day, usually discontinued after the first year) and mycophenolic acid (180–720 mg/8–12 h).

Right ventricular endomyocardial (BEM) biopsies were performed weekly in the first month, every two weeks through the fourth month and monthly through the end of the first year. Thereafter, patients were followed up regularly with a transthoracic echocardiography and an annual coronary angiography.

In the case of severe forms of postoperative dysfunction, initial signs of severe PGF and difficult weaning from cardiopulmonary bypass, the implantation of extracorporeal membrane oxygenators (ECMO) was performed [14,15,16]. As soon as the recovery of cardiac function was observed, a weaning test was completed, and weaning was achieved whenever possible.

### 2.4. Statistical Analysis

Continuous variables are expressed as median and interquartile ranges. Categorical variables are presented as absolute numbers and percentage. The characteristics of SD vs. MD were compared with a Student’s *t* test for continuous variables and with Pearson’s chi-square test or with Fisher’s exact test for categorical values. The significance of the covariates in the univariate analysis of hospital mortality and PGF was also assessed in the same fashion. A multivariable analysis of hospital mortality and PGF was performed by forward conditional logistic regression, including covariates with a univariate *p*-value ≤ 0.20. Univariate and multivariable analyses of mid-term mortality were performed with a Cox regression model (covariates with a univariate *p*-value ≤ 0.20 entered the multivariable analysis). The proportional hazard assumption was verified with the graphical Schoenfeld residuals method. Results are reported as the odds ratio (OR), 95% confidence interval and *p*-value. Survival curves were plotted with the Kaplan–Meier method. Significance was set at *p* < 0.05. Analyses were performed using the R System (R Development Core Team. R: A language and environment for statistical computing. Vienna, Austria: R Foundation for Statistical Computing; 2015).

## 3. Results

A total of 238 patients who underwent orthotopic HTx between January 2012 and December 2020 at the University of Padua were included in the study. Based on the characteristics of the donor, the recipients were divided into two groups: SD recipients (174 patients, 73.1%) and MD recipients (64 patients, 26.9%) (Table 1 and Table 2).

The mean follow-up time was 3.2 ± 2.6 years. The overall Kaplan–Maier (KM) survival rates were 81.2% (76.3%–86.5%) and 79.6% (74.5%–85.1%), respectively, at 1 and 5 years (Figure 1).

### 3.1. MD Recipients

Of 64 MDs, 61 were considered marginal for age over 60 years-old, 2 for focal lesion of the coronary artery, needing coronary artery bypass graft at the time of HTx and 1 for reduced left ventricular performance. Thirty-day and hospital mortality were 14% and 23%, respectively (Table 3). Most of the hospital mortality (73%) was related to multiorgan failure (MOF), mainly as a complication of severe PGF. Acute renal failure requiring continuous venovenous hemofiltration (CVVH) was the most frequent postoperative adverse event (50% of MDs). Severe PGF occurred in 28% of patients. All of these were assisted by ECMO support with a mean time of 4 ± 3 days. In 45% of the cases, weaning from ECMO was achieved. Clinical acute rejection and CAV occurred, respectively, in 24 (37.5%) and 14 (21.9%) recipients during the follow-up (Table 3).

During the univariate analysis for follow-up mortality, a higher preoperative bilirubin level (*p* = 0.08), higher rate of CVVH (*p* = 0.001) and cold ischemic time (*p* = 0.12) resulted as significant risk factors. By the multivariable analysis, only recipients’ characteristics were identified as risk factors [renal impairment (0.95–0.99; *p* = 0.037) and inotropic support (1.17–13.44; *p* = 0.027)]. No donors’ characteristics were significantly associated (Table 4).

During the univariate and multivariable analyses for PGF, cold ischemic time and donor inotropic support were independent risk factors (1.00–1.02; *p* < 0.03 and 1.06–39.23; *p* = 0.04) (Table 5). A logistic regression model showed a linear association between cold ischemic time and PGF (OR 1.01; IC 95% 1.00–1.02; *p* = 0.03) (Figure 2).

### 3.2. SD Recipients versus MDs

The basic characteristics of the SD and MD donors and recipients are summarized in Table 1 and Table 2. MD recipients were older (*p* < 0.001), suffering from a higher incidence of arterial hypertension (*p* = 0.04), chronic obstructive pulmonary disease (*p* = 0.002) and lower glomerular filtration rate (*p* = 0.03). The diagnoses of ischemic cardiomyopathy, primary dilated cardiomyopathy, previous cardiac surgery and left ventricular assist device support were comparable (Table 1). The emergent and urgent states (according to the definitions of the Italian national centers) were also similar (*p* = 0.5), as well as the pretransplant provisional mechanical support (*p* = 0.2).

Marginal donors were older (*p* < 0.001) and had a higher incidence of diabetes (*p* = 0.02). Although not significant, arterial hypertension (*p* = 0.08), hyperlipidemia (*p* = 0.09) and coronary heart disease (*p* = 0.08) were also more frequent in MDs. Smoking as a risk factor was recorded in 28.1% of MDs. The percentage of inotropic support (74% vs. 72%; *p* = 0.6) and the mean cold ischemic time of the graft (207 ± 60 min. vs. 197 ± 58 min.; *p* = 0.2) were similar.

Considering the differences at baseline between the two groups, overall survival was significantly higher in SD than in MD recipients (83.8% vs. 68.1% at 5 years, log-rank = 0.01, Figure 3A). This finding was mainly associated with a higher early mortality rate in MD recipients. However, in the multivariate Cox regression, risk-adjusted mortality for baseline characteristics of MD versus SD recipients [HR 1.8 (0.8–4.1); *p* = 0.2] and 5-year conditional survival (log-rank = 0.6) were comparable (Figure 3A,B).

## 4. Comment

The use of organs from MDs, beyond the controversial opinions, is slowly spreading, to overcome the cardiac donors’ undersupply. In the absence of a universal consensus on the definition of the cardiac risk profiles for the donors and guidelines on the use of organs from high-risk or MDs, the main challenge for the transplant team is to avoid complications on the waiting list and establish the correct timing of HTx [17,18]. As reported by Lietz et al. [19], despite an increase in postoperative mortality and the risk of transplant-related coronary heart disease, “older grafts” bring more benefits than staying on the waiting list. However, the models for risk stratification in heart failure patients awaiting HTx and the mid long-term outcomes of MDs still appear to be limited [20,21].

In the current study, a cohort of MDs was first analyzed at a 5-year interval. Early and late survival were satisfactory, as well as the rate of adverse events. Hospital mortality in the MD recipient cohort was 23%, with an estimated 1 and 5-year survival of 70% and 69%, respectively. A multivariate analysis of MD recipients showed that decreased renal function and increased inotropic support of recipients were associated with higher hospital mortality. Univariate and multivariable analyses of overall mortality did not provide further significant results. Inotropic support and end-stage renal disease have been already reported as independent predictors of mortality [22]. Indeed, increased inotropic exposure may affect HTx survival in terms of myocardial dysfunction, poor outcome and long-term complications [23]. As described by Trivedi et al. [24], risk factors of the recipients are more relevant in predicting post-transplant survival than donor factors. Our analysis confirmed these results. We suggest for appropriate matching, in the case of long-distance organ procurement, to not use recipients with high inotropic support and end-stage renal disease. However, there is no objective scoring system that independently uses recipient and donor risk factors to predict post-HTx survival [25].

The most common cause of hospital mortality of MD recipients is PGF, although several causes can be considered responsible [25]. This occurs in up to 30% of MD recipients [26]. In our study, inotropic donors and prolonged ischemic time were found to be the main risk factors. A statistically significant and linear association was observed between cold ischemic time and the development of PGF. An option to overcome the limitations of cold ischemic storage is ex-vivo normothermic perfusion (e.g., organ care system (OCS) (Transmedics, Andover, MA, USA), which allows one to monitor the function of the graft, addressing the time constraints crucial for geographic allocation [27]. In fact, the evaluation of the lactate trend can be useful in evaluating myocardial protection of the graft and in the recognition of unsuitable organs. The largest study on the use of OCS in the preservation and evaluation of non-standard donor hearts is the EXPAND-Heart-Trial [28]. The results of this study showed excellent short-term posttransplant results, particularly a low PGF rate, supporting ex-vivo perfusion as a procedure to be used for MD procurement. However, the results observed with OCS were not superior to the standard cold storage procedure, concluding that further investigation was mandatory. Our current experience with OCS and MDs is limited to five cases, so we cannot draw any conclusions from this. However, the results of our study on MDs can be a driving force in the more frequent OCS use.

In the second part of our study, we then compared the mid-term outcome of recipients from MDs versus SDs. Although the perioperative mortality of MD recipients appeared to be higher than that of SDs, mid-term conditional survival was comparable, emphasizing adequate graft function in the long term. Several reports showed no inferiority of marginal donors, even though the criteria for defining MDs were less selective than ours (age > 55 years, cold ischemic time > 4 h, high dose of noradrenaline) [8,9,18].

However, in selected scenarios, HTx was performed using hearts affected by coronary arteries and valvular disease, or congenital heart defects. This group of organs can be successfully transplanted with a concomitant procedure at the time of HTx [29].

A low incidence of acute and chronic rejection events in MD recipients was found in our population, and no significant differences were observed when comparing them to SD recipients. Conversely, we observed a higher risk of infection as reported by Sugimara et al. They retrospectively analyzed the clinical outcome of HTx recipients focusing on the donor’s left ventricular function. Although prolonged posttransplant recovery was observed, no difference in terms of 1-year survival was shown [30].

Commonly, the development of CAV is considered a non-negligible complication in the use of MDs [9]. However, this was not found to be significantly different in our cohorts.

### Study Limitations

This is a single-center retrospective study. Although the number of our study population can be considered relatively small, our marginal donor pool represents one-third of the total MDs assigned in our country during the study period. A multi-center effort is undoubtedly needed to expand the population and further validate our preliminary results. Unfortunately, we could not perform a comparison between the cold storage procurement strategy and the ex-vivo perfusion procedure. Therefore, further investigation is needed on this aspect. Finally, it was not possible to standardize and include in the analysis the biological age of donors and recipients. Therefore, only chronological age was included in the analysis [31].

## 5. Conclusions

The use of marginal donors could be considered a valid resource in an era of cardiac donor organ shortages. Provided that careful donor–recipient matching occurs, satisfactory early and long-term results are achievable.

The use of MDs after prolonged ischemic times, increased inotropic support of the MD or the recipient and decreased renal function are associated with worse outcomes in terms of both mortality and incidence of PGF.

## Figures and Tables

**Figure 1 jcm-11-02665-f001:**
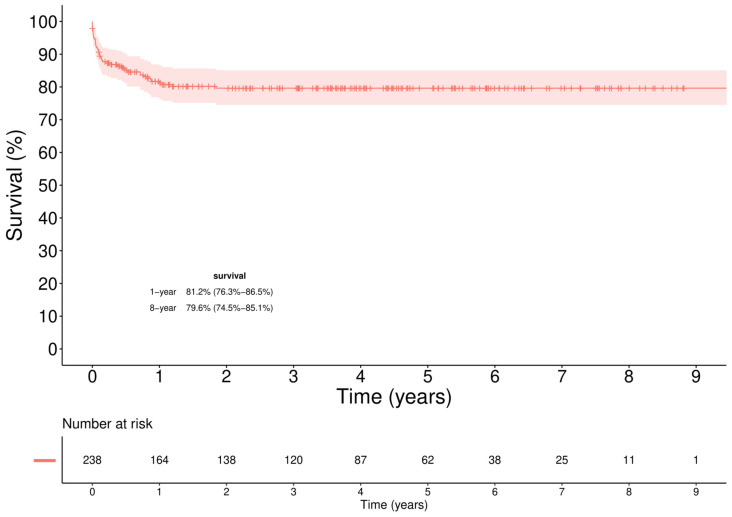
Overall survival in HTx population. Survival at 1 year was 81.2% (76.3–86.5%), at 8 years 79.6% (74.5–85.1%).

**Figure 2 jcm-11-02665-f002:**
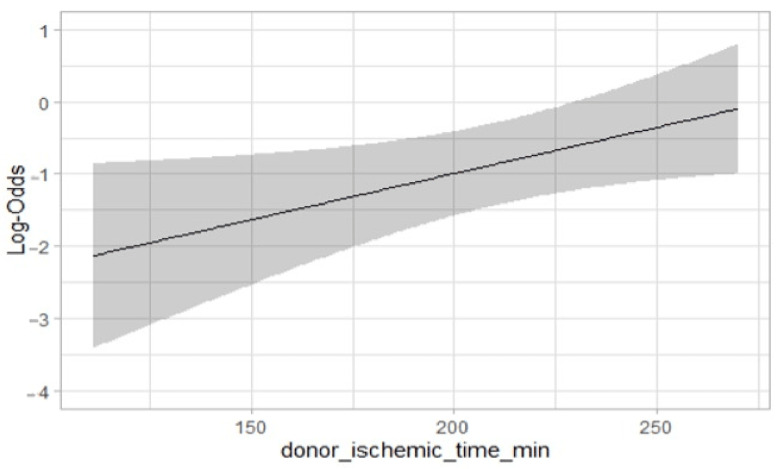
Graphical representation of linear and significant association between cold ischemic time in MDs and primary graft failure onset. (Logistic regression model).

**Figure 3 jcm-11-02665-f003:**
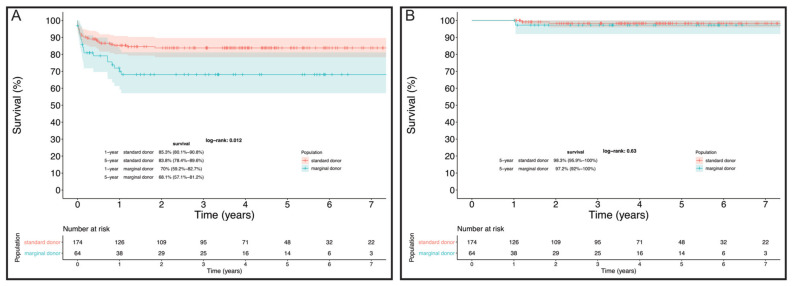
(**A**) Overall survival in SDs compared to MDs; (**B**) 5-year conditional survival of MDs compared to SDs.

**Table 1 jcm-11-02665-t001:** (a,b) Basic characteristics of organ recipients from standard versus marginal donors.

	SDs (*n* = 174)	MDs (*n* = 64)	*p*
	*n* (%) or Median (IQR)	*n* (%) or Median (IQR)	
**(a)**			
**Gender (Female)**	39 (22.4%)	17 (26.6%)	0.5
**Age (years)**	56.4 (46.8–63.3)	63.6 (55.3–66.8)	**<0.001**
**Cardiac diagnosis**			0.5
**Dilatative**	67 (38.5%)	18 (28.1%)	
**Ischemic**	71 (40.8%)	33 (51.6%)	
**Congenital**	12 (6.9%)	3 (4.7%)	
**Valvular**	3 (1.7%)	2 (3.1%)	
**Hypertrophic**	10 (5.7%)	2 (3.1%)	
**Other**	11 (6.3%)	6 (9.4%)	
**BSA (m^2^)**	1.7 (1.7–2.0)	1.8 (1.7–1.9)	0.6
**Dyslipidemia**	63 (36.2%)	25 (39.1%)	0.8
**Hypertension**	62 (35.6%)	33 (51.6%)	**0.04**
**Cancer**	9 (5.2%)	2 (3.1%)	0.7
**Diabetes**	31 (17.8%)	16 (25.0%)	0.3
**PVD**	12 (6.9%)	7 (10.9%)	0.2
**COPD**	10 (5.7%)	13 (20.3%)	**0.002**
**ICD**	129 (74.1%)	54 (84.4%)	0.1
**Cerebral event**	28 (16.1%)	16 (25.0%)	0.1
**Smoker**	63 (36.2%)	23 (35.9%)	0.6
**Genetic syndrome**	11 (6.3%)	3 (4.7%)	0.8
**Previous cardiac surgery**	67 (38.5%)	32 (50.0%)	0.1
**Bilirubin (µmol/L)**	14.6 (9.3–22.6)	15.0 (7.2–22.7)	0.3
**GFR (mL/min/mq)**	71.0 (49.0–89.0)	51.0 (44.0–68.0)	**0.03**
**(b)**			
**Intracorporeal LVAD**	49 (28.2%)	19 (29.7%)	
**Length of LVAD support (months)**	13.1 (4.8–31.6)	19.0 (13.7–28.0)	0.4
**List status**			**0.02**
**2B**	74 (42.5%)	28 (43.8%)	
**2A**	51 (29.3%)	21 (32.8%)	
**1**	5 (2.9%)	7 (10.9%)	
**HU**	44 (25.3%)	8 (12.5%)	
**Status 1 + HU**	49 (28.2%)	15 (23.4%)	0.5
**Waiting list time (months)**	4.8 (1.1–18.1)	6.5 (2.2–21.6)	0.5
**ICU-stay**	62 (35.6%)	15 (23.4%)	0.2
**Inotropic support**	47 (27.0%)	15 (23.4%)	0.6
**Mechanical ventilation**	7 (4.0%)	4 (6.3%)	0.5
**CVVH**	8 (4.6%)	3 (4.7%)	0.9
**Temporary-MCS**	42 (24.14%)	8 (12.5%)	0.2
**IABP**	1 (0.6%)	0 (0.0%)	0.9
**ECMO**	18 (10.3%)	3 (4.7%)	0.2
**Paracorporeal-LVAD**	12 (6.9%)	3 (4.7%)	0.8
**Paracorporeal-RVAD**	3 (1.7%)	1 (1.6%)	0.8
**Paracorporeal-BiVAD**	8 (4.6%)	1 (1.6%)	0.5

Abbreviations: BSA, body surface area; COPD, chronic obstructive pulmonary disease; CVVH, continuous venovenous hemofiltration; ECMO, extracorporeal membrane oxygenator; GFR, glomerular filtration rate; HU, high urgency; IABP, intra-aortic balloon pump; ICD, implantable cardioverter defibrillator; ICU, intensive care unit; LVAD, left ventricular assistance device; MCS, mechanical circulatory support; PVD, peripheral vascular disease, RVAD, right ventricular assist device.

**Table 2 jcm-11-02665-t002:** Basic characteristics of standard versus marginal donors.

	Standard (*n* = 174)	Marginal (*n* = 64)	*p*
	*n* (%) or Median (IQR)	*n* (%) or Median (IQR)	
**Age (years)**	45.0 (29.5–53.0)	64.0 (62.0–66.0)	**<0.001**
**Match age**	−10.0 (−23.3–1.2)	2.8 (−1.2–10.1)	**<0.001**
**Gender (female)**	71 (40.8%)	33 (51.6%)	0.2
**Mismatch gender**	55 (31.6%)	20 (31.3%)	0.9
**Inotropic support**	130 (74.7%)	46 (71.9%)	0.6
**Cardiac arrest**	33 (19.0%)	9 (14.1%)	0.4
**Cath abnormalities**	1 (0.6%)	4 (6.3%)	0.08
**Hypertension**	17 (9.8%)	12 (18.8%)	0.08
**Smoker**	51 (29.3%)	18 (28.1%)	0.9
**Dyslipidemia**	3 (1.7%)	4 (6.3%)	0.1
**Diabetes**	1 (0.6%)	4 (6.3%)	**0.02**
**Cold ischemic time (minutes)**	220.0 (160.0–250.0)	200.0 (156.3–234.3)	0.2

**Table 3 jcm-11-02665-t003:** Outcomes and complications of organ recipients from standard versus marginal donors.

	SDs (*n* = 174)	MDs (*n* = 64)	*p*
	*n* (%) or Median (IQR)	*n* (%) or Median (IQR)	
**Severe PGF**	36 (20.7%)	18 (28.1%)	0.2
**Postoperative ECMO support (days)**	5.0 (3.0–6.5)	3.5 (2.0–6.8)	0.2
**CVVH**	54 (31.0%)	32 (50.0%)	**0.01**
**Intrahospital infection**	61 (35.1%)	30 (46.9%)	0.1
**Clinical cellular rejection**	62 (35.6%)	24 (37.5%)	0.9
**CAV**	28 (16.1%)	14 (21.9%)	0.6
**Cerebral event**	17 (9.8%)	12 (18.8%)	0.07
**30-day mortality**	13 (7.5%)	9 (14.1%)	0.1
**Hospital mortality**	21 (12.1%)	15 (23.4%)	**0.04**
**Cause of hospital mortality**			
**MOF**	10 (47.6%)	11 (73.3%)	
**Neurologic**	2 (9.5%)	1 (6.7%)	
**Infection**	4 (19.1%)	3 (20.0%)	
**Other**	5 (23.8%)	0 (0.0%)	

Abbreviations: CAV, cardiac graft vasculopathy; CVVH, continuous venovenous hemofiltration; ECMO, extracorporeal membrane oxygenator; MOF: multiorgan failure; PGF, primary graft failure.

**Table 4 jcm-11-02665-t004:** Univariate and multivariate analysis of organ recipients from marginal donors for mortality.

Recipient Characteristics	MDs Alive (*n* = 44)	MDs Mortality (*n* = 19)	*p*	Multivariate Analysis
	*n* (%) or Median (IQR)	*n* (%) or Median (IQR)		
**Listing status**			0.12	
**2B**	24 (55%)	3 (16%)		
**2A**	12 (27%)	9 (47%)		
**1**	3 (7%)	4 (21%)		
**HU**	5 (11%)	3 (16%)		
**Peripheral vascular disease**	2 (5%)	4 (21%)	0.01	
**Platelets count (10^3^/mm^3^)**	187 (167–249)	189 (128–255)	0.13	
**Bilirubin (µmol/L)**	13 (7–21)	20 (10–44)	0.08	
**C-reactive Protein (mg/L)**	3.8 (2.9–15.1)	26 (4–99)	0.12	
**eGFR (mL/min/m^2^)**	60 (49–84)	47 (42–53)	0.02	**HR 0.98 (0.95–0.99; *p* = 0.04)**
**Inotropic support**	5 (11%)	10 (53%)	<0.01	**HR 3.96 (1.17–13.44; *p* = 0.03)**
**CVVH**	0 (0%)	3 (16%)	<0.01	
**Mechanical ventilation**	1 (2%)	3 (16%)	0.08	
**ICU stay**	8 (18%)	7 (37%)	0.13	
**Paracorporeal-LVAD**	1 (2%)	2 (11%)	0.17	
**Donors Characteristics**	**MDs Alive (*n* = 44)**	**MDs Mortality (*n* = 19)**	** *p* **	**Multivariate Analysis**
	***n* (%) or Median (IQR)**	***n* (%) or Median (IQR)**		
**Cold ischemic time (min)**	200 (149–234)	216 (169–240)	0.12	
**Hypertension**	6 (14%)	6 (32%)	0.13	
**Donor/Recipient BSA**	0.0 (−0.2–0.1)	0.1 (0.0–0.2)	0.12	

Abbreviations: CVVH, continuous venovenous hemofiltration; GFR, glomerular filtration rate; HU, high urgency; BSA, body surface area; ICU, intensive care unit; LVAD, left ventricular assistance device.

**Table 5 jcm-11-02665-t005:** Univariate and multivariate analysis of organ recipients from marginal donors for primary graft failure.

Recipients’ Characteristics	Marginal Donors w/o PGF (*n* = 46)	Marginal Donors with PGF (*n* = 18)	*p*	Multivariate Analysis
	*n* (%) or Median (IQR)	*n* (%) or Median (IQR)		
**List Status**			0.02	
**2B**	23 (50.0%)	5 (27.8%)		
**2A**	12 (26.1%)	9 (50.0%)		
**1**	3 (6.5%)	4 (22.2%)		
**HU**	8 (17.4%)	0 (0.0%)		
**Status 1 + HU**	11 (23.9%)	4 (22.2%)	0.9	
**Donor Characteristics**	**Marginal Donors w/o PGF (*n* = 46)**	**Marginal Donors with PGF (*n* = 18)**	** *p* **	**Multivariate Analysis**
	***n* (%) or Median (IQR)**	***n* (%) or Median (IQR)**		
**Inotropic support**	30 (65.2%)	16 (88.9%)	0.06	**OR 6.50 (1.06–39.22; *p* = 0.04)**
**Cold ischemic time (minutes)**	193.0 (147.3–228.5)	227.5 (180.0–246.0)	0.02	**OR 1.01 (1.00–1.02; *p* = 0.03**

PGF, primary graft failure; HU, high urgency.

## Data Availability

The data that support the findings of this study are available on request from the corresponding author, [TB]. The data are not publicly available due to restrictions, and their information could compromise the privacy of research participants.

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
