# Peer review of "Marginal versus Standard Donors in Heart Transplantation: Proper Selection Means Heart Transplant Benefit"

_jcm, 2022, doi:10.3390/jcm11092665_

Round 1

Reviewer 1 Report

The authors compared the operative results of patients who had heart transplantations with marginal or standard donor hearts. They concluded that proper selection means heart transplant benefit.

I agree their conclusions including the expectation in possibility of OCS.  However there are some questions as below.

  1. P7L101; MDs were defined with the following criteria; a. age over 60 years, b. reduced LV performance(EF 40-50%), c. LV hypertrophy(septal thickness>14mm),d. focal lesion of the coronary artery, e. significant valvular heart disease.; I could not find any details of the a-e groups. For proper selection of the donor, they should mention about at least number of a-d groups. If possible, they should comment which definition is most important for good selection.
  2. P11L171; At univariate analysis for follow-up mortality, higher preoperative bilirubin level (p=0.049): in which P number is different from P=0.08 in Table 4 , higher rate of CVVH(p=0.001): CVVH is not in Table 4 even it is listed in abbreviation of Table 4 (CVVH = renal replacement therapy (P<0.01))?, and cold ischemic time (of donor heart)(P=0.041) : in which P number is different from p=0.12 in Table 4.  

Reviewer 2 Report

Abstract: the abstract is well written and provides all necessary informations.

Introduction: i would suggest to mention primary and secondary endpoints and the design  of the study (retrospective observational) in the introduction.

M&M: I would suggest to move Table 1 to the Results- section. Additionally, I would recommend to mention the listing- system and its implications on urgency in the M&M- section, as not everybody knows the italian graduation. I find Table 1 hard to read- maybe you should divide it into baseline demographic informations and transplant- related informations, such as listing status, VAD, etc.

Please mention the technique used for HTx in your intstitution.

Results: Please use HR consistently.

Discussion: Please check the actual literature- there are some groups examining outcomes after HTx and MD (i.e., Thoracic Cardiovasc Surg . 2021 Sep;69(6):490-496., Clin Transpl 2020 Nov;34(11):e14057., J Card Surg, 2021 Dec;36(12):4828-4829.) and include these in the actual discussion.

Round 2

Reviewer 1 Report

I enjoyed this manuscript very much. The authors revised it according to the reviewers' comments well. 

Only one spelling mistake;L322 : Sugimara →Sugimura